# Strain Range Dependent Cyclic Hardening of 08Ch18N10T Stainless Steel—Experiments and Simulations

**DOI:** 10.3390/ma12244243

**Published:** 2019-12-17

**Authors:** Jaromír Fumfera, Radim Halama, Radek Procházka, Petr Gál, Miroslav Španiel

**Affiliations:** 1Department of Mechanics, Biomechanics and Mechatronics, Faculty of Mechanical Engineering, Czech Technical University in Prague, Technicka 4, 16000 Prague 6, Czech Republic; jaromir.fumfera@fs.cvut.cz (J.F.); miroslav.spaniel@fs.cvut.cz (M.Š.); 2Department of Applied Mechanics, Faculty of Mechanical Engineering, VSB-Technical University of Ostrava, 17.listopadu 2172/15, 70800 Ostrava, Czech Republic; Petr.Gal@ujv.cz; 3Comtes FHT, a. s., Průmyslová 995, 334 41 Dobřany, Czech Republic; rprochazka@comtesfht.cz; 4ÚJV Řež, a. s., Hlavní 130, Řež, 250 68 Husinec, Czech Republic

**Keywords:** austenitic steel 08Ch18N10T, cyclic plasticity, cyclic hardening, experiments, finite element method, low-cycle fatigue

## Abstract

This paper describes and presents an experimental program of low-cycle fatigue tests of austenitic stainless steel 08Ch18N10T at room temperature. The low-cycle tests include uniaxial and torsional tests for various specimen geometries and for a vast range of strain amplitude. The experimental data was used to validate the proposed cyclic plasticity model for predicting the strain-range dependent behavior of austenitic steels. The proposed model uses a virtual back-stress variable corresponding to a cyclically stable material under strain control. This internal variable is defined by means of a memory surface introduced in the stress space. The linear isotropic hardening rule is also superposed. A modification is presented that enables the cyclic hardening response of 08Ch18N10T to be simulated correctly under torsional loading conditions. A comparison is made between the real experimental results and the numerical simulation results, demonstrating the robustness of the proposed cyclic plasticity model.

## 1. Introduction

Austenitic stainless steels, for example, 316L in PWR (pressurized water reactor) and 08Ch18N10T in the Russian VVER concept (water–water power reactor), are usually used for components in primary circuit reactor internals (a block consisting of guided tubes, a core barrel, a core barrel bottom and a core shroud), in main primary pipes, and so forth. During their design life, these components must withstand mechanical operational loads (e.g., pressure pulses and vibrations), thermal loads (regimes such as heating up and shut-downs), corrosive loads and also irradiation. These regimes subject the reactor internals to cyclic loading.

When designing or assessing the long term operation of existing structural components, it is necessary to include fatigue evaluations. In the last decade, the finite element method (FEM) with phenomenological models has mainly been used in practical applications [1]. A description and a short history of the development of constitutive models of cyclic plasticity has been provided by the authors in a previous publication [2]. Their goal is to describe as accurately as possible the stress-strain behavior of the material, which is found on the basis of experiments under cyclic loading conditions [3]. A small deviation in the stress-strain prediction can lead to a major fatigue error, especially in low cycle fatigue. In this case, stainless steels show cyclic hardening in the initial stage, followed by cyclic softening [4,5]. This phenomenon depends on the strain range and also on the type of loading. Non-proportional loading induces more cyclic hardening than proportional loadings. The most sensitive materials are materials with low stacking fault energy, for example, austenitic stainless steels [6]. Low-cycle fatigue tests of this type were presented for example, by Jin et al. in Reference [7]. They presented results for 316L stainless steel under proportional and non-proportional loadings. In another study, Xing et al. [8] presented the results of experimental testing on 316L stainless steel under proportional and non-proportional loadings with various strain amplitudes. The authors also presented a numerical study and compared the numerical results with the experimental data. They used the visco-plastic numerical model, based on the Ohno-Wang kinematic hardening rule.

The temperature in VVER concept reactor usually does not exceed 350 ${}^{\circ }C$ in most components. Temperature effect have significant influence, which is presented in Reference [9]. The additional hardening due to non-proportional loading has been investigated by many authors. The basic concept involves modifying the isotropic or kinematic hardening rule with a non-proportional parameter. For example, Benallal and Marquis [10] introduced the non-proportional angle, which is defined as the angle between the direction of the increment in plastic deformation and the direction of the deviatoric stress. Another approach was introduced by Tanaka [11]. He introduced the fourth rank tensor, which characterizes the internal dislocation structure of the material. This parameter is dependent on the loading path.

The goal of all the studies mentioned above was to understand the behavior of the material under specific cyclic loading conditions and to provide the material data for a better fatigue and lifetime assessment of the structural parts. This paper follows up on the main author’s previous paper [2] which presents some results of uniaxial low-cycle fatigue tests of austenitic stainless steel 08Ch18N10T at room temperature. The experimental program includes uniaxial tests of hourglass-type specimens and is now extended by new results for notched specimens with 3 different notch geometries considering strain amplitudes up to 3%. Torsional loading tests of notched-tube specimens are also newly presented.

In a previous paper [2], the authors presented a new constitutive material model that is used for finite element (FE) simulations of experiments on 08Ch18N10T material. The constitutive material model is based on the Chaboche model. The proposed material model is in very good agreement with uniaxial loading condition results. In this paper, the model has been modified to provide a better description of the torsional loading. This modification also enables the cyclic hardening response of 08Ch18N10T steel to be simulated correctly under torsional loading conditions. The constitutive material model is based on the memory surface introduced in the stress space, which is analogous to the theory of Jiang and Sehitoglu [12] for treating the impact of the strain amplitude on the stress response of the material. The new theory is shown on the kinematic hardening rule based on Chaboche’s model with three backstress parts. Recently, an approach has been introduced that takes into account a new internal variable called virtual backstress, corresponding to a cyclically stable material. This provides an easy way to identify the parameters and to reduce the number of material parameters. A comparison between the real experimental results and the numerical simulation results demonstrates the robustness of the constitutive plasticity model.

## 2. Experiments

The experimental section describes the low-cycle fatigue test measurements of specimens in pure tension/compression mode and in torsion mode.

### 2.1. Experimental Setup

Pure axial tension-compression tests were carried out using a MAYES electromechanical testing machine with a loading capacity of 100kN. The test specimens were placed in MTS 646 hydraulic collet grips to ensure repeatability of the alignment conditions in tensile/compression mode. The axial deformation of the specimens was controlled by an MTS 634.25 extensometer with an initial gage length of 10 mm with a 50% measuring range, for uniform gage specimens and with an initial length of 20 mm with a 20% measuring range, for elliptically-shaped specimens.

Pure torsion tests were conducted on an MTS Bionix servo-hydraulic testing machine with an axial load capacity of 25kN and 250Nm in torsion. The test specimens were carefully mounted in MTS 647 hydraulic wedge grips and were tested with the axial load control set to zero. An EPSILON 3550 axial/torsional extensometer was employed to measure and control the torsional shear angle with a range of $\pm 2{}^{\circ }$. The initial gauge length of the extensometer was 25mm. The whole test setup is shown in Figure 1.

These tests were conducted at room temperature and were loaded using a triangular waveform at a strain rate of $0.002\;s^{-1}$. During the experimental measurements all channels were recorded, for example, time, force/torque, displacement/angle, axial/torsional extensometer, with a recording frequency of 20Hz.

The digital image correlation (DIC) was used for an analysis of the 3D deformation on the surface of some specimens, see Figure 1. During cyclic loading, the frame rate was set to cover at least 20fps per one loading cycle. The MERCURY RT optical measuring system was used to capture and analyze the 3D images. The configuration of the system consists of two 5Mpx CMOS BONITO cameras with circular polarizing filters to reduce the glare from the reflected surface of the specimen.

The test setup (see Figure 1) on the MTS servo-hydraulic testing machine consists of hydraulic wedge grips, a notched specimen and an EPSILON axial/torsional extensometer and a snapshot of a notched specimen (see Figure 1) under a loading with a random contrast pattern, which the DIC algorithm requires and a strain map on the surface.

The stochastic pattern on the surface of the specimen and two digital video cameras allows 3D strain measurements throughout the fatigue life until fracture, with resolution of 1100 DPI (1 px = 0.22 μm). In addition, the DIC system can continuously store all captured images in the computer memory. The fatigue life of each loading condition takes at least several dozen of cycles, even tens of thousands of cycles, which can generate up to hundreds of thousands of images to be processed. All captured images were processed later in post processing to prevent data loss. This loss occurs when the bitrate increases while real-time processing is being used.

### 2.2. Experimental Program

The experimental program consists of 6 series of specimens. The first series is used for the material parameter identification process. According to the ASTM E606 standard [13], the classic uniform-gage geometry of a specimen is limited to a total strain amplitude of $\epsilon_{a}$ = 0.5%. For higher strain levels, non-uniform hour-glass type geometry is required in order to prevent buckling. The material parameters identification series (IDF) was therefore compiled from uniform-gage (UG) specimens (see Figure 2) and non-uniform-gage specimens with an elliptical longitudinal section (E9, see Figure 2). To identify the material parameters (described in detail in Reference [2]), it is necessary to know the stress-strain curves in the cycles. For UG specimen geometry, tested according to Reference [13], this can be calculated directly from the elongation of the extensometer and from the force measured during the experiment. For E9 specimen geometry, the strain was measured by the DIC (due to the experimental setup, the strain cannot be calculated directly from the elongation of the extensometer for non-uniform gage geometries).

The next series consists of E9 geometry (see Figure 2), notch geometry with an R=1.2mm (R1.2, see Figure 3), geometry with an R=2.5mm notch (R2.5, see Figure 3) and geometry with an R=5mm notch (R5, see Figure 4). The last series is the notched tube geometry (NT, see Figure 4), which was exposed to torsional loading.

All boundary conditions of the experiments and their simulations are together with resulting experimental lifetimes reported in Appendix A.

## 3. Constitutive Model with Strain Range Dependency

The concept of single yield surface plasticity with strain range dependency is used. Isothermal conditions are considered, since the influence of the strain rate is not taken into account. The constitutive model is described in detail in Reference [2], so just a brief recapitulation of some key equations is presented here.

### 3.1. Cyclic Plasticity and Memory Surface

The plasticity function is defined as
(1)$$\begin{array}{c} f=\sqrt{\frac{2}{3}\left(s-a\right):\left(s-a\right)}-Y=0, \end{array}$$
where s is the deviatoric part of stress tensor σ, a is the deviatoric part of back-stress α. The actual yield surface size *Y* is defined as
(2)$$\begin{array}{c} Y=\sigma_{y}+R, \end{array}$$
where *R* is the isotropic variable and $\sigma_{Y}$ is the initial size of the yield surface. The accumulated plastic strain increment dp is defined as
(3)$$\begin{array}{c} dp=\sqrt{\frac{2}{3}d\epsilon^{p}:d\epsilon^{p}}. \end{array}$$

The superposition of the virtual back stress parts is defined as
(4)$$\begin{array}{c} \alpha_{virt}=\sum_{i=1}^{M}\alpha_{virt}^{i}, \end{array}$$
and for each part
(5)$$\begin{array}{c} d\alpha_{virt}^{i}=\frac{2}{3}C_{i}d\epsilon_{p}-\gamma_{i}\alpha_{virt}^{i}dp. \end{array}$$

For 08Ch18N10T material, three backstress parts are taken into consideration, so M=3.

The evolution of the memory surface size $R_{M}$ is directed by the following rule
(6)$$\begin{array}{c} dR_{M}=H\left(g\right)\langle L:d\alpha_{virt}\rangle \end{array}$$
where
(7)$$\begin{array}{c} g=∥\alpha_{virt}∥-R_{M}<=0 \end{array}$$
and
(8)$$\begin{array}{c} L=\frac{\alpha_{virt}}{∥\alpha_{virt}∥}. \end{array}$$

### 3.2. Isotropic Hardening

The cyclic isotropic hardening is linear in *p*, defined incrementally as
(9)$$\begin{array}{c} dR=R_{0}\left(R_{M}\right)dp, \end{array}$$
where
(10)$$\begin{array}{c} R_{0}\left(R_{M}\right)=A_{R}R_{M}^{2}+BR_{M}+C_{R}\;\mathrm{for}\;R_{M}\geq R_{M0} \end{array}$$
(11)$$\begin{array}{c} R_{0}\left(R_{M}\right)=A_{R}R_{M0}^{2}+BR_{M0}+C_{R}\;\mathrm{otherwise}, \end{array}$$
where $A_{R}$, $B_{R}$, $C_{R}$ and $R_{M0}$ are material parameters.

### 3.3. Kinematic Hardening

Chaboche’s kinematic hardening rule is used in this study. The backstress is composed of *M* parts
(12)$$\begin{array}{c} \alpha =\sum_{i=1}^{M}\alpha_{i}, \end{array}$$
the memory term is a function of memory surface $R_{M}$ and accumulated plastic strain *p*
(13)$$\begin{array}{c} d\alpha_{i}=\frac{2}{3}C_{i}d\epsilon_{p}-\gamma_{i}\phi \left(p,R_{M}\right)\alpha_{i}dp, \end{array}$$
where *M*, $C_{i}$ and $\gamma_{i}$ are the same as in Equation (Equation 5). Function ϕ is defined as
(14)$$\begin{array}{c} \phi \left(p,R_{M}\right)=\phi_{0}+\phi_{cyc}\left(p,R_{M}\right), \end{array}$$
where $\phi_{0}$ is a material parameter. $\phi_{cyc}$ is a function defined as follows
(15)$$\begin{array}{c} d\phi_{cyc}=\omega \left(R_{M}\right)\;\cdot \;\left(\phi_{\infty }+\phi_{cyc}\left(p,R_{M}\right)\right)dp \end{array}$$
(16)$$\begin{array}{c} \phi_{\infty }\left(R_{M}\right)=A_{\infty }R_{M}^{4}+B_{\infty }R_{M}^{3}+C_{\infty }R_{M}^{2}+D_{\infty }R_{M}+E_{\infty } \end{array}$$
(17)$$\begin{array}{c} \omega \left(R_{M}\right)=A_{\omega }+B_{\omega }R_{M}^{-C_{\omega }}\;\mathrm{for}\;R_{M}\geq R_{M\omega } \end{array}$$
(18)$$\begin{array}{c} \omega \left(R_{M}\right)=A_{\omega }+B_{\omega }R_{M\omega }^{-C_{\omega }}\;\mathrm{otherwise} \end{array}$$
where $A_{\infty }$, $B_{\infty }$, $C_{\infty }$, $D_{\infty }$, $E_{\infty }$, $A_{\omega }$, $B_{\omega }$, $C_{\omega }$ and $R_{M\omega }$ are material parameters.

### 3.4. Modification for Torsional Loading

The original plasticity model shows very good prediction under uniaxial loading conditions [2]. The original model also predicts well for notched specimen geometries but produces an error of up to about 15% under shear stress loading conditions, as will be shown in Section 6. For a low loading level (see Figure 5a)), where there is limited cyclic hardening, the prediction of the original model [2] is satisfactory. For a high loading level (see Figure 5b)), the model overpredicts the cyclic hardening under dominant shear stress loading conditions and the formulation of the material model needs to be modified.

The first modification of the original model [2] is to separate the memory surface function into two memory surfaces. Memory surface $R_{M}$ for the isotropic hardening part remains the same as in the original model defined by the set of Equations (Equation 4)–(Equation 8). The new memory surface $R_{M\phi }$ for the kinematic hardening part is modified and is defined by analogy as
(19)$$\begin{array}{c} \alpha_{virt\phi }=\sum_{i=1}^{M}\alpha_{virt\phi }^{i} \end{array}$$
(20)$$\begin{array}{c} d\alpha_{virt\phi }^{i}=\frac{2}{3}C_{i}d\epsilon_{p}-\gamma_{i}K\alpha_{virt\phi }^{i}dp, \end{array}$$
where
(21)$$\begin{array}{c} K=(\delta_{IJ}+\left(1-\delta_{IJ}\right)K_{shear}), \end{array}$$
where $\delta_{IJ}$ is Kronecker delta, *I*, *J* are indexes of stress tensor σ and $K_{shear}$ is a new material parameter. The rest of the equations for defining the memory surface of the kinematic hardening part remain analogous to the original model [2]:(22)$$\begin{array}{c} dR_{M\phi }=H\left(g_{\phi }\right)\langle L_{\phi }:d\alpha_{virt\phi }\rangle \end{array}$$
where
(23)$$\begin{array}{c} g_{\phi }=∥\alpha_{virt\phi }∥-R_{M\phi }<=0 \end{array}$$
and
(24)$$\begin{array}{c} L_{\phi }=\frac{\alpha_{virt\phi }}{∥\alpha_{virt\phi }∥}. \end{array}$$

A quick analysis of this modified formulation shows that it provides practically the same prediction in uniaxial loading conditions (because $R_{M\phi }≃R_{M}$) as the original formulation. However, depending on the value of $K_{shear}$, it can give a different prediction under shear loading conditions: it is more effective for higher loading levels than for lower loading levels and it can reduce the over prediction of the model for $K_{shear}>1$.

The second modification to the original model [2], also associated with the memory surface, is to omit limits $R_{M\omega }$ and $R_{M0}$ and to set boundaries of the memory surfaces instead: $R_{M}^{min}$ and $R_{M}^{max}$. The value of the memory surface $R_{M}$ and $R_{M\phi }$ used for controlling the isotropic and kinematic hardening part can lie only between these two bounds. For simplification and for mathematically correct expression, the memory surface size that is used, $R_{M}^{used}$, is defined as
(25)$$\begin{array}{ccc} R_{M}^{used} & = & R_{M}^{min}\;\mathrm{for}\;R_{M}<R_{M}^{min} \end{array}$$
(26)$$\begin{array}{ccc} R_{M}^{used} & = & R_{M}\;\mathrm{for}\;R_{M}^{min}<R_{M}<R_{M}^{max} \end{array}$$
(27)$$\begin{array}{ccc} R_{M}^{used} & = & R_{M}^{max}\;\mathrm{for}\;R_{M}>R_{M}^{max} \end{array}$$
and analogously for $R_{M\phi }^{used}$. The variable $R_{M}$ in Equations (Equation 13)–(Equation 18) of the original model is simply replaced by variable $R_{M\phi }^{used}$. The modified form of the kinematic hardening equations is now
(28)$$\begin{array}{c} d\alpha_{i}=\frac{2}{3}C_{i}d\epsilon_{p}-\gamma_{i}\phi \left(p,R_{M\phi }^{used}\right)\alpha_{i}dp \end{array}$$
(29)$$\begin{array}{c} \phi \left(p,R_{M\phi }^{used}\right)=\phi_{0}+\phi_{cyc}\left(p,R_{M\phi }^{used}\right) \end{array}$$
(30)$$\begin{array}{c} d\phi_{cyc}=\omega \left(R_{M\phi }^{used}\right)\;\cdot \;\left(\phi_{\infty }+\phi_{cyc}\left(p,R_{M\phi }^{used}\right)\right)dp \end{array}$$
(31)$$\phi_{\infty }\left(R_{M\phi }^{used}\right)=A_{\infty }{\left(R_{M\phi }^{used}\right)}^{4}+B_{\infty }{\left(R_{M\phi }^{used}\right)}^{3}+C_{\infty }{\left(R_{M\phi }^{used}\right)}^{2}+D_{\infty }R_{M\phi }^{used}+E_{\infty }$$
(32)$$\begin{array}{c} \omega \left(R_{M\phi }^{used}\right)=A_{\omega }+B_{\omega }{\left(R_{M\phi }^{used}\right)}^{-C_{\omega }}. \end{array}$$

The third modification of the original model [2] is the definition of the formulation of isotropic hardening as a non-linear formulation in *p* as
(33)$$\begin{array}{c} dR=A_{R}\;\cdot \;\exp\left(B_{R}\;\cdot \;R_{M}^{used}\right)\;\cdot \;p^{C_{R}}, \end{array}$$
where $A_{R}$, $B_{R}$ and $C_{R}$ are material parameters. This very important modification deserve a short analysis. In the original model [2], for cyclic loading, the actual yield stress *Y* increases practically linearly with the number of cycles. This means that with many cycles, the actual yield stress *Y* can theoretically go higher than the total stress amplitude and the computed deformation becomes only elastic.

## 4. Identification of Material Parameters

The material parameter identification process for 08Ch18N10T is based on knowing the shape of the stress-strain hysteresis loops during the fatigue life. A total of twelve uniaxial specimens and eight torsional specimens are used for the identification process. This is described in detail in References [14] and [2], so just a brief recapitulation of the key steps updated by the unique features of the proposed modification to the material model is done here.

The Young modulus *E*, the Poisson ratio μ and the yield strength $\sigma_{y}$ are obtained from a tensile test. The actual yield strength evolution during the fatigue life is determined using the root mean square error method. Chaboche material parameters $C_{1}$, $\gamma_{1}$, $C_{2}$, $\gamma_{2}$, $C_{3}$, $\gamma_{3}$ are identified from two selected hysteresis loops (the bigger loop and the smaller loop).

The first guess of the memory surface size $R_{M}$ for each specimen is computed. It is assumed here that $R_{M\phi }≃R_{M}$. Boundary parameters $R_{M}^{min}$ and $R_{M}^{max}$ are simply the maximum and minimum values of $R_{M}$ computed in the identification process.

The actual yield stress is fitted as a function of $R_{M}^{used}$ and parameters $A_{R}$, $B_{R}$, $C_{R}$ are found from Equation (Equation 33).

Using the experimental data from the tensile test and performing a simulation of this test, parameter $\phi_{0}$ is found based on the Equation (Equation 13) as an optimal value of ϕ. The value of function ϕ from Equation (Equation 13) is found, using a similar optimization process as for determining the Chaboche material parameters. $\phi_{\infty }$ is the value of ϕ for $n=N_{d}$, where $N_{d}$ is the number of cycles after which the crack occurs on the specimen and the force starts to drop during the experiment. From Equation (Equation 16), $\phi_{\infty }$ is then set as a function of $R_{M}^{used}$ by finding material parameters $A_{\infty }$, $B_{\infty }$, $C_{\infty }$, $D_{\infty }$, $E_{\infty }$.

For each NT geometry specimen tested, the Error value in each cycle between the experimental amplitude of torque $T_{a\;exp}$ and the simulation amplitude of torque $T_{a\;sim}$ can be defined as
(34)$$\begin{array}{c} Error=\left(T_{a\;exp}-T_{a\;sim}\right)/T_{a\;exp}\;\cdot \;100\left[\%\right]\;. \end{array}$$

The MeanError over all cycles is calculated as
(35)$$\begin{array}{c} MeanError=\frac{1}{N_{d}}\sum_{n=1}^{N_{d}}Error_{n}, \end{array}$$
where index *n* is the number of cycles. The total error over all NT geometry specimens tested is defined as
(36)$$\begin{array}{c} TotalError=\frac{1}{S}\sum_{s=1}^{S}MeanError_{s}, \end{array}$$
where *s* is the NT specimen index and S=8 is the total number of NT specimens tested (see Table A3 in Appendix A for details).

For the different $K_{shear}$ from Equation (Equation 21), the TotalError value is captured in Figure 6. The final $K_{shear}$ material parameter is identified as the optimal value of $K_{shear}$ where the TotalError is minimal.

The material value parameters are presented in Table 1.

The experimental data from the IDF series of experiments can also be plotted into fatigue diagram $\epsilon_{a}$-$N_{f}$, where $N_{f}$ is the number of cycles to failure and $\epsilon_{a}$ is the amplitude of the total strain. Due to the experimental setup, $\epsilon_{a}$ is not completely constant during the experiment in the case of E9 geometry (during the experiments, the amplitude of extensometer elongation $\frac{\Delta L_{ext}}{2}$ is controlled to be constant, so for UG geometry the $\epsilon_{a}$ is also constant but it is not completely constant for E9 geometry), so the mean value during the experiment is plotted. Fatigue data are shown in Figure 7. Other lifetimes are reported in the form of tabular data in Appendix A.

## 5. FE Simulations

The geometry of most specimens is not uniform, so the non-uniform stress and strain field in their cross-section are expected and FEA must be used for simulations. The constitutive model is implemented into Abaqus FE software using the USDFLD subroutine. FE models of each of the tested geometries were created, see Figure 8, Figure 9 and Figure 10. The symmetry boundary condition is defined on the right edge of the model. The left edge of the model always corresponds with the cross-section where the extensometer is attached to the body of the specimen during the experiment. The displacement boundary condition on the upper edge of the FE model is created with the same amplitude value as was recorded from the extensometer during the experiment. Abaqus CAX8R mesh elements are used for the axisymmetric models and C3D8R elements are used for the NT geometry, which is a 3D model with cyclic symmetry. The element size in fine mesh areas has been determined using sensitivity study to 0.1mm.

The Abaqus Chaboche plasticity material model with combined hardening and the USDFLD subroutine is used. The equations of the constitutive model are coded into the USDFLD subroutine for calculating the actual memory surfaces size $R_{M}^{used}$ and $R_{M\phi }^{used}$, which, combined with the accumulated plastic strain *p*, determines the actual yield stress *Y*, the value of function ϕ and the memory term of the Chaboche model $\phi \;\cdot \;\gamma_{i}$. The full Abaqus USDFLD subroutine code written in Fortran is available in Appendix B.

This subroutine makes possible to use the material model presented here in engineering computations. Combined with the material parameters identification process described in Section 4, it can also be used for other materials.

## 6. Experimental and Simulation Results

The implementation of the plasticity model presented here (including the presented modification) into FE code was verified using FE simulations of all experiments mentioned in Section 2.2. The following figures show some results of experiments and their FE simulations. Due to the large scale of the experimental program, only two representative specimens with low and high load levels were selected for demonstration in this section. The results of remaining specimens are presented in the form of error values in following tables. The compared variables in each figure are the amplitudes of the force measured during the experiment ($F_{a\;exp}$) and computed by the FE simulations ($F_{a\;sim}$). Two constitutive models are shown—the original model [2] and the modified model presented in this paper. The actual error between each FE simulation and the experiment and the mean error value, are also displayed. 

The error between the experiment and the FE simulation in each cycle *n* is calculated simply as
(37)$$\begin{array}{c} Error=\frac{F_{a\;exp}-F_{a\;sim}}{F_{a\;exp}}\times 100\%. \end{array}$$

The mean error and the total error are calculated using Equations (Equation 35) and (Equation 36) considering corresponding number of specimens in the series.

The Figure 11 and Table 2 show the experimental and simulation results of E9 geometry series representing the uniaxial loading conditions. The prediction capability of these two models is comparable.

The NT geometry series results are in Figure 12 and Table 3. In this case, the compared variables are the amplitudes of the torque measured during the experiment ($T_{a\;exp}$) and computed by the FE simulations ($T_{a\;sim}$). The errors are calculated using Equations (Equation 34)–(Equation 36). For this geometry, the difference in the prediction capability of the original model and the modified model is not the same—the modified model provides a better prediction of the cyclic hardening of the material under torsional loading for high loading levels.

Finally, the notched specimen geometry series R1.2, R2.5 and R5 follows on Figure 13, Figure 14 and Figure 15 and Table 4, Table 5 and Table 6. The stress field in the cross-section of these specimens is no longer uniaxial and the prediction capabilities of both models are also comparable.

## 7. Discussion

As has been shown in the previous sections, the model proposed in Reference [2] can capture very well the static and cyclic stress-strain curve for uniaxial loading conditions with a reasonable number of material parameters. Using the modification proposed in this paper, with only two extra material parameters, the error under torsional loading conditions can be reduced significantly, without any harm under uniaxial loading conditions, as is shown in Table 7, where the total errors (defined in Equation (Equation 36)) are summarized. Both models also produce very good predictions for notched specimen geometries, where the stress-strain field is not uniaxial. 

In Figure 11, Figure 12, Figure 13, Figure 14 and Figure 15, the range of response quantity axis has been chosen to make visible the difference between experimental values and predicted ones. That is why the error seems to be higher than actually is. This is true especially in case of the lowest strain amplitude. 

In the calibration process dealing with torsional loading, the $K_{shear}$ value is found as a compromise between all loading levels, so the proposed modification improves prediction for most, but not all, specimens tested (see Table 3 and Table 7).

It should be pointed out that presented tests consist only of single loading modes. Combinations of these modes, for example, proportional and also non-proportional combination of tension and torsion probably induces cyclic non-proportional hardening and another cyclic phenomenons. These conditions are also limiting for eventual FE analysis on the real components. Combined loading conditions considering proportional as well as non-proportional loading are potential topics for future investigation.

## 8. Conclusions

This paper has described the experimental setup and the experimental program for a low-cycle fatigue test of 08Ch18N10T austenitic stainless steel. Using FE simulations, material model [2] capable of capturing the strain-range dependent cyclic hardening has been newly verified on notched specimens, where the stress-strain field is non-uniform and for torsional loading. With a newly proposed modification, model can correctly simulate cyclic hardening also for shear stress loading conditions.

The extensive experimental program was subsequently completely simulated. The Chaboche plasticity model combined with non-linear isotropic hardening has already been implemented into Abaqus commercial FE software. The model presented here can easily be implemented into Abaqus using the USDFLD subroutine as a simple extension of the Abaqus default cyclic plasticity model. The full Fortran code of subroutine can be found in Appendix B. This implementation makes the proposed model ready to use for some engineering computations. The usage limitations are given by the conditions under which the model has been tested (simple, uncombined loading).

The original cyclic plasticity model presented in [2] provides a good prediction of the cyclic response of uniaxial and notched specimens. With the modification for torsional loading that has been presented here, it can also provide a good prediction of cyclic hardening under torsional loading conditions. It can also easily be applied to the Abdel-Karim-Ohno model or to a modified version with promised ratcheting prediction [8]. The model can be extended by standard techniques for use in the area of viscoplasticity [15]. 

The calibration of the cyclic plasticity model was described briefly in this paper and was used with experimental data available for 08Ch18N10T. In future work, an automated process for identifying material parameters could be prepared in a similar way as in [16]. Some authors of this paper also work on the material parameters identification using results from DIC measurements in order to reduce number of necessary specimens for recently expensive technologies of 3D printing of metals [17]. 

## Figures and Tables

**Figure 1 materials-12-04243-f001:**
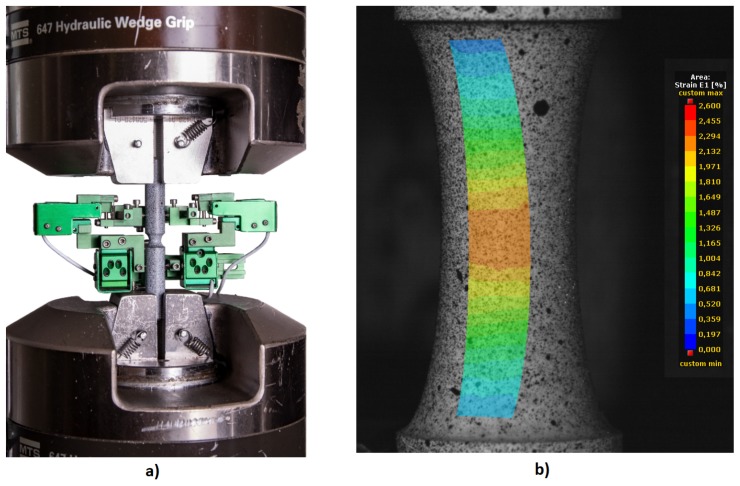
Experiment: (**a**) Experimental Setup, (**b**) digital image correlation (DIC) Snapshot of Specimen IDF-6.

**Figure 2 materials-12-04243-f002:**
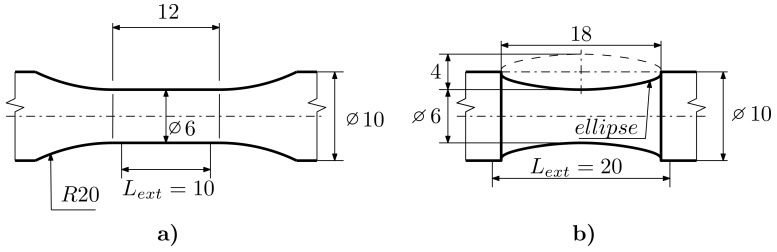
IDF Specimen Geometry: (**a**) UG, (**b**) E9.

**Figure 3 materials-12-04243-f003:**
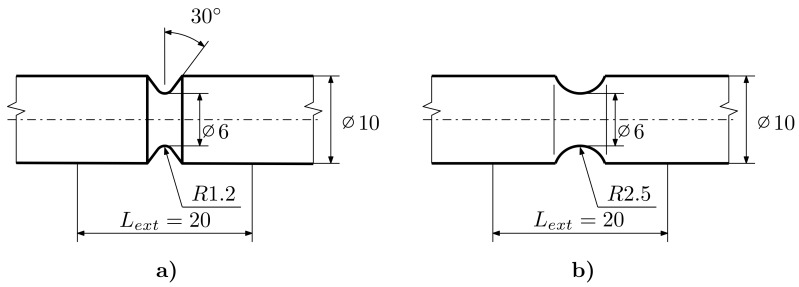
Notched Specimens: (**a**) R1.5, (**b**) R2.5.

**Figure 4 materials-12-04243-f004:**
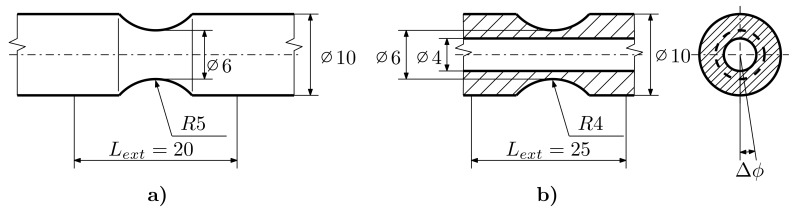
Notched and Notched Tube Specimens: (**a**) R5, (**b**) NT.

**Figure 5 materials-12-04243-f005:**
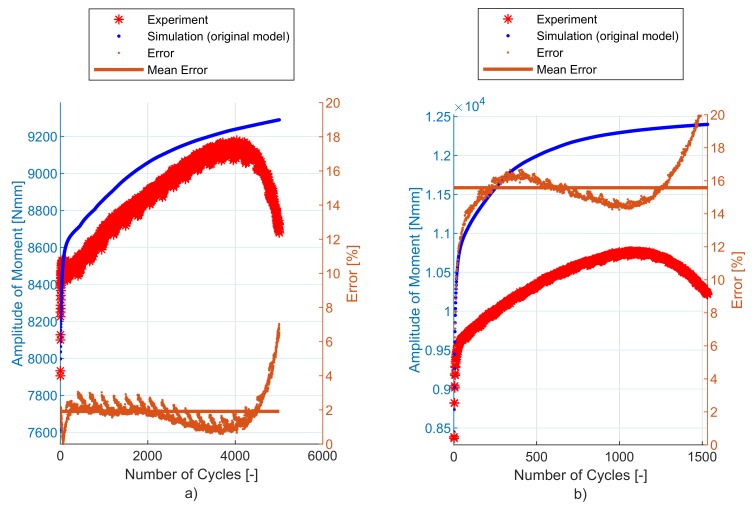
Original model under torsional loading: (**a**) specimen NT-1 (low loading level), (**b**) specimen NT-6 (high loading level).

**Figure 6 materials-12-04243-f006:**
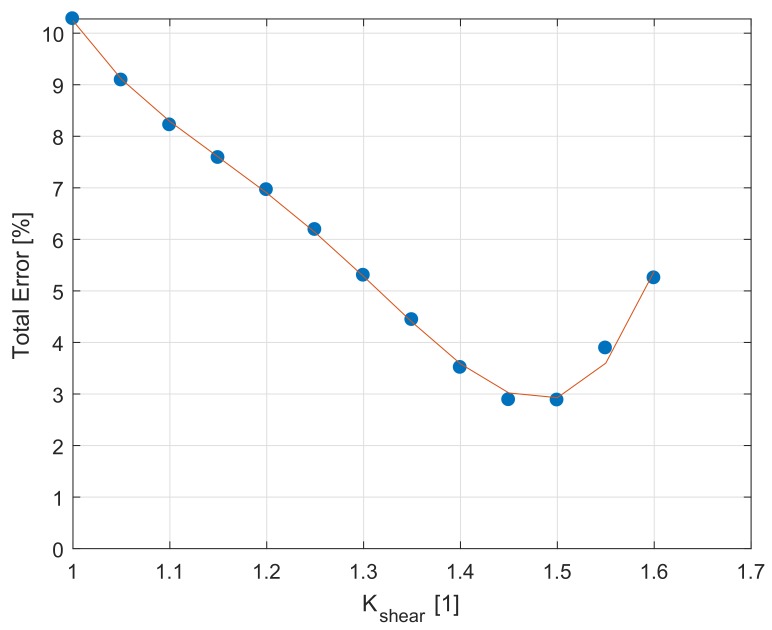
Identification of material parameter $K_{shear}$.

**Figure 7 materials-12-04243-f007:**
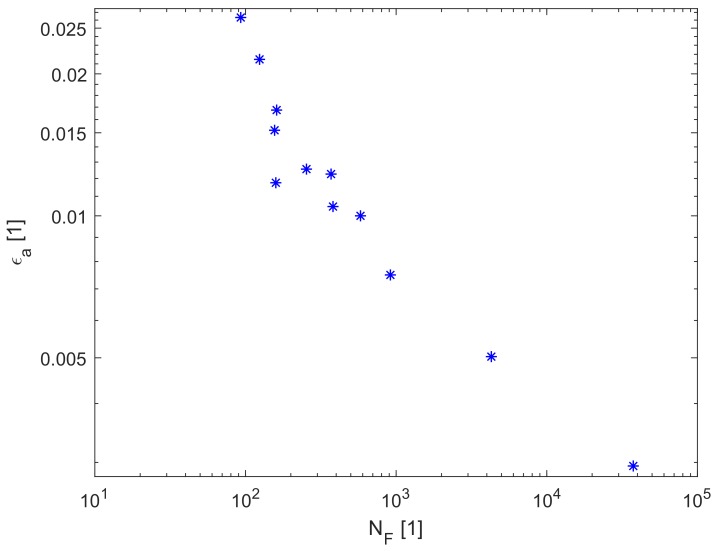
Fatigue data of the IDF series of experiments.

**Figure 8 materials-12-04243-f008:**
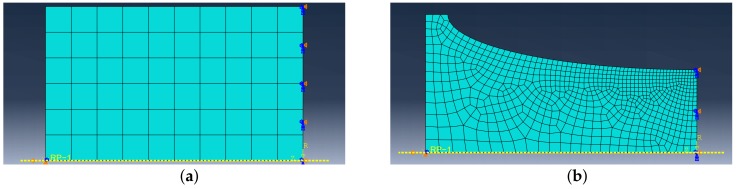
FE model: (**a**) UG, (**b**) E9.

**Figure 9 materials-12-04243-f009:**
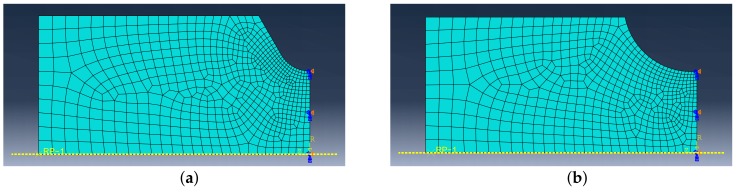
FE model: (**a**) R1.2, (**b**) R2.5.

**Figure 10 materials-12-04243-f010:**
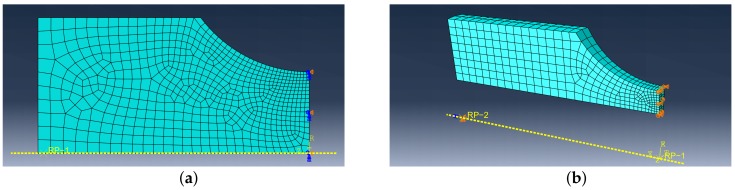
FE model: (**a**) R5, (**b**) NT.

**Figure 11 materials-12-04243-f011:**
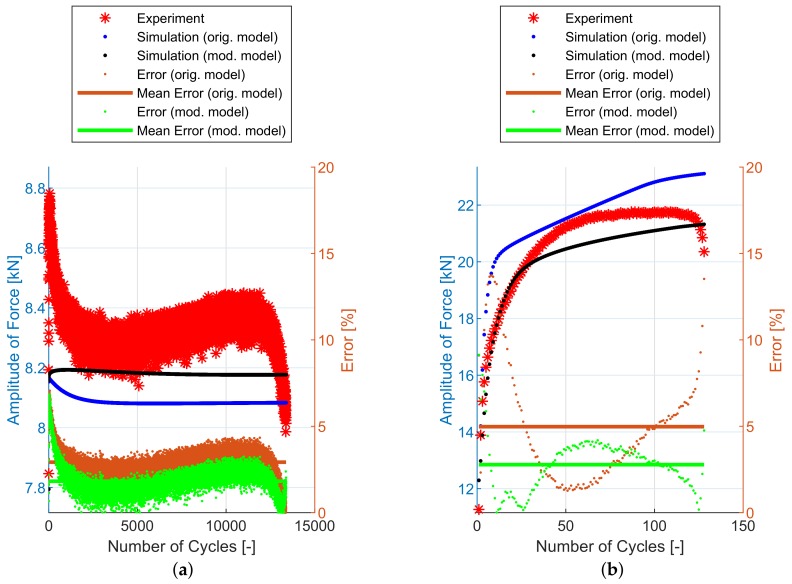
Amplitude of force—experiment vs. simulations [2]: (**a**) E9-1, (**b**) E9-17.

**Figure 12 materials-12-04243-f012:**
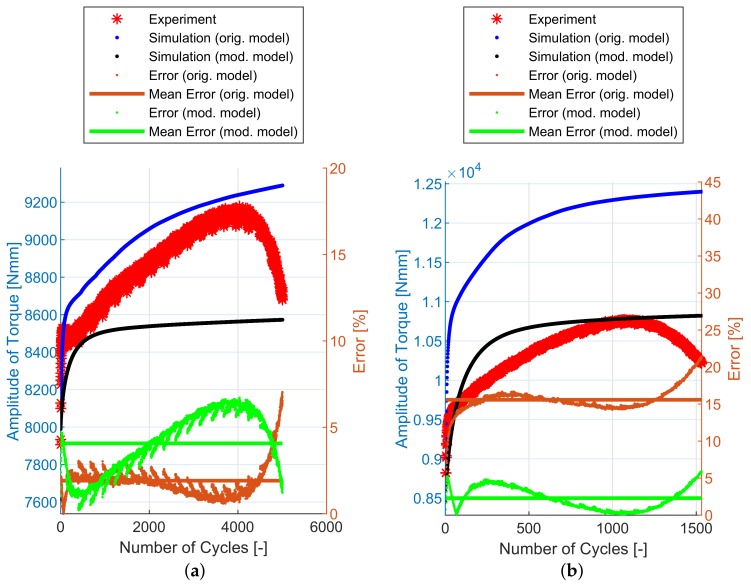
Amplitude of force—experiment vs. simulations: (**a**) NT-1, (**b**) NT-6.

**Figure 13 materials-12-04243-f013:**
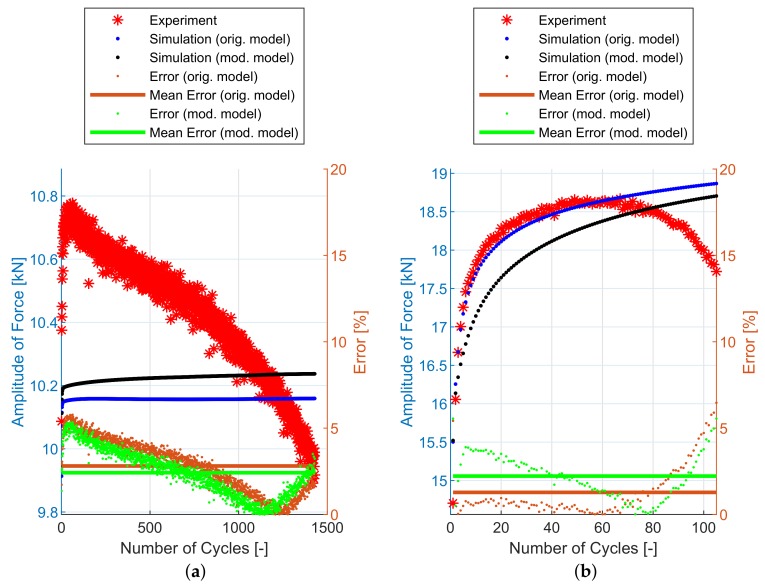
Amplitude of force—experiment vs. simulations: (**a**) R1.2-1, (**b**) R1.2-18.

**Figure 14 materials-12-04243-f014:**
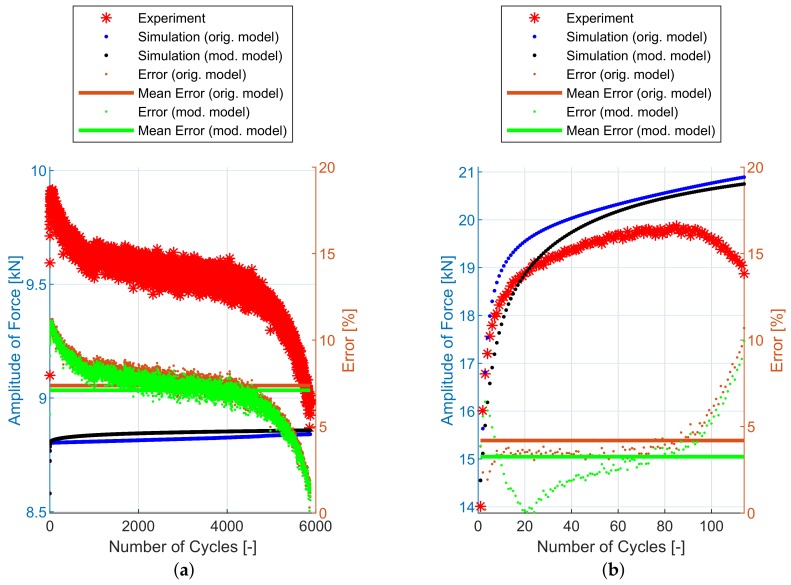
Amplitude of force—experiment vs. simulations: (**a**) R2.5-1, (**b**) R2.5-21.

**Figure 15 materials-12-04243-f015:**
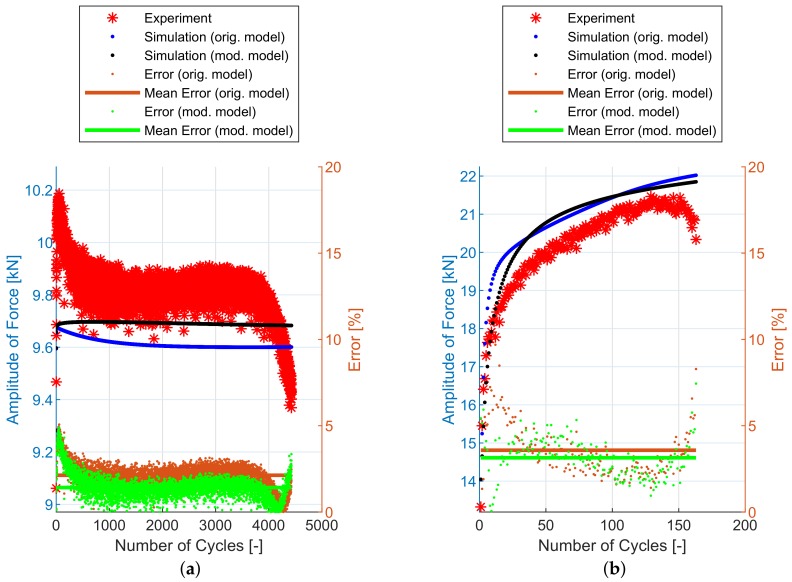
Amplitude of force—experiment vs. simulations: (**a**) R5-1, (**b**) R5-24.

**Table 1 materials-12-04243-t001:** Material parameters of the new proposed model for 08Ch18N10T.

E[MPa]	ν	$\sigma_{y}\;\left[\mathrm{MPa}\right]$	$C_{1}$	$\gamma_{1}$	$C_{2}\;\left[\mathrm{MPa}\right]$
210,000	0.3	150	63,400	148.6	10,000
$\gamma_{2}$	$C_{3}\;\left[\mathrm{MPa}\right]$	$\gamma_{3}$	$A_{\infty }$	$B_{\infty }$	$C_{\infty }$
911.4	2000	0	$-1.3127\times 10^{-9}$	$1.7981\times 10^{-6}$	$-8.6705\times 10^{-4}$
$D_{\infty }$	$F_{\infty }$	$A_{R}\;\left[{\mathrm{MPa}}^{-1}\right]$	$B_{R}$	$C_{R}\;\left[\mathrm{MPa}\right]$	$R_{M}^{\min}\;\left[\mathrm{MPa}\right]$
$1.6678\times 10^{-1}$	−10.600	$3.0113\times 10^{-1}$	$1.4865\times 10^{-1}$	$1.1818\times 10^{-2}$	130.54
$A_{\omega }$	$B_{\omega }$	$C_{\omega }$	$R_{M}^{\max}\;\left[\mathrm{MPa}\right]$	$\phi_{0}$	$K_{\mathrm{shear}}$
0	$2.0024\times 10^{-13}$	−4.8591	506.59	2.3178	1.5

**Table 2 materials-12-04243-t002:** Mean error of all E9 specimens tested—experiment vs. simulations [2].

Specimen Name	Orig. Model Mean Err. [%]	Mod. Model Mean Err. [%]	Specimen Name	Orig. Model Mean Err. [%]	Mod. Model Mean Err. [%]
E9-1	2.9226	1.8207	E9-10	7.8144	8.8757
E9-2	2.3311	1.2756	E9-11	2.5028	3.9003
E9-3	2.4027	1.1938	E9-12	4.3523	6.5915
E9-4	1.6977	0.7773	E9-13	4.0929	3.4343
E9-5	8.0687	7.0447	E9-14	2.1610	3.8515
E9-6	8.8658	7.4521	E9-15	2.9195	2.9485
E9-7	11.7310	10.4229	E9-16	1.8601	2.7524
E9-8	3.8241	3.9171	E9-17	4.9766	2.7579
E9-9	9.8245	9.5508			

**Table 3 materials-12-04243-t003:** Mean error of all NT specimens tested—experiment vs. simulations.

Specimen Name	Orig. Model Mean Err. [%]	Mod. Model Mean Err. [%]	Specimen Name	Orig. Model Mean Err. [%]	Mod. Model Mean Err. [%]
NT-1	1.9100	4.0682	NT-5	14.2137	1.3947
NT-2	0.8367	5.9823	NT-6	15.5549	2.2815
NT-3	11.2048	1.3797	NT-7	13.1168	1.5014
NT-4	11.1021	1.0934	NT-8	8.8054	4.7887

**Table 4 materials-12-04243-t004:** Mean error of all R1.2 specimens tested—experiment vs. simulations.

Specimen Name	Orig. Model Mean Err. [%]	Mod. Model Mean Err. [%]	Specimen Name	Orig. Model Mean Err. [%]	Mod. Model Mean Err. [%]
R1.2-1	2.8075	2.4172	R1.2-10	1.6518	1.7538
R1.2-2	3.7011	3.1679	R1.2-11	2.0827	2.2332
R1.2-3	2.2438	2.2027	R1.2-12	3.9411	3.2028
R1.2-4	2.8530	2.7056	R1.2-13	2.5308	3.1540
R1.2-5	2.8984	2.7105	R1.2-14	1.4521	1.8444
R1.2-6	4.7877	4.4405	R1.2-15	3.6781	2.6435
R1.2-7	1.4888	1.4897	R1.2-16	1.5820	1.9106
R1.2-8	7.1382	6.7943	R1.2-17	1.6089	2.5930
R1.2-9	2.4171	2.2355	R1.2-18	1.2789	2.2219

**Table 5 materials-12-04243-t005:** Mean error of all R2.5 specimens tested—experiment vs. simulations.

Specimen Name	Orig. Model Mean Err. [%]	Mod. Model Mean Err. [%]	Specimen Name	Orig. Model Mean Err. [%]	Mod. Model Mean Err. [%]
R2.5-1	7.3714	7.1025	R2.5-12	2.1944	1.6489
R2.5-2	8.1586	7.6327	R2.5-13	1.2466	1.0057
R2.5-3	9.1468	8.6587	R2.5-14	8.7778	9.1473
R2.5-4	6.8139	6.8130	R2.5-15	2.6624	3.0678
R2.5-5	6.6714	6.6118	R2.5-16	1.4643	1.3563
R2.5-6	9.9838	9.1708	R2.5-17	0.9873	1.5697
R2.5-7	4.3249	3.4860	R2.5-18	1.4020	1.4515
R2.5-8	3.8551	3.8250	R2.5-19	1.6099	2.6423
R2.5-9	1.0034	0.9027	R2.5-20	0.9634	2.4069
R2.5-10	4.7921	4.9816	R2.5-21	4.1944	3.2605
R2.5-11	1.9673	2.1464			

**Table 6 materials-12-04243-t006:** Mean error of all R5 specimens tested—experiment vs. simulations.

Specimen Name	Orig. Model Mean Err. [%]	Mod. Model Mean Err. [%]	Specimen Name	Orig. Model Mean Err. [%]	Mod. Model Mean Err. [%]
R5-1	2.1303	1.4186	R5-13	6.7479	6.8700
R5-2	2.0673	1.8112	R5-14	5.1055	5.4414
R5-3	0.7021	0.8284	R5-15	1.3043	1.4251
R5-4	0.9757	0.9284	R5-16	1.1829	1.3661
R5-5	1.4847	1.4209	R5-17	3.6903	3.6048
R5-6	1.7435	1.6993	R5-18	3.1399	2.9518
R5-7	2.9066	2.7548	R5-19	6.1649	6.1226
R5-8	5.3372	5.4106	R5-20	2.8263	2.6683
R5-9	4.9004	4.5530	R5-21	1.0485	1.2882
R5-10	2.3623	2.6227	R5-22	8.2167	7.6119
R5-11	7.0110	6.8065	R5-23	2.2011	1.6441
R5-12	2.3912	3.1025	R5-24	3.5803	3.1425

**Table 7 materials-12-04243-t007:** Total error comparison between the original model and the modified model.

Geometry	The Original Nodel [2] Total Error [%]	The Modified Model Total Error [%]
E9	4.84	4.61
NT	9.60	2.85
R1.2	2.79	2.76
R2.5	4.27	4.23
R5	3.30	3.23

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
