# Peer review of "Strain Range Dependent Cyclic Hardening of 08Ch18N10T Stainless Steel—Experiments and Simulations"

_materials, 2019, doi:10.3390/ma12244243_

Round 1

Reviewer 1 Report

This work presents a study of strain hardening for stainless steel with different strain range through experiments and FEA simulation. The topic is suitable for the journal. Good agreements between experiments and simulations were reported in the manuscript. The reviewer suggests a revision for authors to address the following issues.
1) The significance and contribution of the current work should be further discussed. Why FEA is necessary if the experimental method is available and easily implementable? For example, the authors could discuss the optimization of cyclic hardening with FEA and optimization algorithm. Therefore, the usefulness of the FEA model could be better understood by the readers. Please review the following references related to optimization and inverse analysis.https://doi.org/10.1007/s00170-018-2508-6; etc.
2) The reference should be removed from the abstract. In addition, please check the grammar and spelling throughout the manuscript.
3) There were many results presented in the manuscript and appendices. Please present them in a better way that the readers can visualize the results in terms of FEA model accuracy and strain hardening effect. For example, plot two figures for the FEA model accuracy and strain hardening results for all testing conditions.
4) The literature review is inadequate without reviewing the commonly use constitutive models and their corresponding applications, such as the Johnson-Cook model, and Zerilli-Amstrong model, plasticity-law model. Please review the following references. https://doi.org/10.3390/ma12020284; etc.
5) What are the limitations of the current work especially the FEA model? What are the future works for improving prediction accuracy and computational efficiency?
6) Again, the significance and contribution of the current work should be emphasized in the conclusion. Therefore, novelty and usefulness in real applications should be discussed.

Reviewer 2 Report

An interesting topic and methods used for the research. However, the presentation of methods and results is a mess. There should be shorter and clearer presentation of methods and results both in formulas and graphs shown in the results section. In other words, authors should keep only few mathematical expressions and few graphs which show the core scientific contribution of the present work.

Reviewer 3 Report

This paper presents findings from an experimental program that was carried to investigate the low-cycle fatigue tests of austenitic stainless steel 08Ch18N10T via uniaxial and torsional tests. This paper is interesting as it provides good insights into how austenitic stainless steel 08Ch18N10T behave under cyclic loading and it also derives an improved material model. The following items are to be addressed:

As the authors have stated, austenitic stainless steel is often used in reactors and hence its properties and behavior under a combination of loading (high temperature etc.) are of interest. What was the rationale behind carrying out the provided tests given that they were conducted at ambient conditions? Similar to the above point, the authors are encouraged to strengthen the introduction section by referring to some of the work that examine properties of austenitic stainless steel under elevated temperature. Some of these can be the following and the authors may use these references and/or others: https://doi.org/10.1016/j.conbuildmat.2019.04.182 https://doi.org/10.1061/(ASCE)MT.1943-5533.0002842 doi:10.4995/ASCCS2018.2018.7011 Was the coupons tested according to a specific standard? Please state the standard in the text. Was the DIC equipment calibrated after each test? Or only one time at the beginning of the testing program? 5 (as well as 19) show that results from simulation does not seem to match that in tests towards failure. Comment and further discuss this. How was element size arrived at? Was is based on a sensitivity study? The discussion section is very short. This needs to be improved and further articulated. Overall, it seems that the authors have done a good amount of work in their previous studies. The novelty (and need) for this paper are to be highlighted and justified.

Round 2

Reviewer 1 Report

The authors have addressed the raised issues. The reviewer has no further concern regarding this work. It can now be accepted for publication.

Reviewer 3 Report

Thanks for your effort.